# Plasma Treatment of Large-Area Polymer Substrates for the Enhanced Adhesion of UV–Digital Printing

**DOI:** 10.3390/nano14050426

**Published:** 2024-02-26

**Authors:** Michal Fleischer, Zlata Kelar Tučeková, Oleksandr Galmiz, Eva Baťková, Tomáš Plšek, Tatiana Kolářová, Dušan Kováčik, Jakub Kelar

**Affiliations:** Department of Plasma Physics and Technology, CEPLANT—R&D Centre for Plasma and Nanotechnology Surface Modifications, Faculty of Science, Masaryk University, Kotlářská 2, 611 37 Brno, Czech Republic; michal.fleischer@mail.muni.cz (M.F.); o.galmiz@mail.muni.cz (O.G.); 484056@mail.muni.cz (E.B.); 461281@mail.muni.cz (T.P.); t.zahoranova@yahoo.com (T.K.); dusan.kovacik@mail.muni.cz (D.K.); jakub.kelar@mail.muni.cz (J.K.)

**Keywords:** UV–digital printing, transparent polymers, low-temperature plasma, surface functionalization, ink adhesion

## Abstract

UV–digital printing belongs to the commonly used method for custom large-area substrate decoration. Despite low surface energy and adhesion, transparent polymer materials, such as polymethylmethacrylate (PMMA) and polycarbonate (PC), represent an ideal substrate for such purposes. The diffuse coplanar surface barrier discharge (DCSBD) in a novel compact configuration was used for substrate activation to improve ink adhesion to the polymer surface. This industrially applicable version of DCSBD was prepared, tested, and successfully implemented for the UV–digital printing process. Furthermore, wettability and surface free energy measurement, X-ray photoelectron spectroscopy, atomic force, and scanning electron microscopy evaluated the surface chemistry and morphology changes. The changes in the adhesion of the surface and of ink were analyzed by a peel-force and a crosscut test, respectively. A short plasma treatment (1–5 s) enhanced the substrate’s properties of PMMA and PC while providing the pre-treatment suitable for further in-line UV–digital printing. Furthermore, we did not observe damage of or significant change in roughness affecting the substrate’s initial transparency.

## 1. Introduction

Plasmas operating at atmospheric pressure have found widespread commercial use as a tool for polymer pre-treatment [1,2]. Activating the polymer surface improves surface free energy, adhesive bonding strength, paint adhesion, and dye uptake [3,4]. Meanwhile, an increase in the material surface roughness can benefit the bonding and coating of the material surface [5]. Atmospheric low-temperature plasmas (LTPs) are an interesting alternative to other pre-treatment methods (e.g., low-pressure plasmas [6,7] or wet chemical or radiation treatment [4,8]) because of their relatively low costs, in-line process capabilities, and personal and environmental safety requirements [9,10,11,12]. However, common atmospheric pressure LTPs are often characterized by properties such as a small treatment area (corona, plasma jet, and pencil), inhomogeneity (gliding arc), requirement of gas flow or an opposite electrode placed under the substrate (corona or volume dielectric barrier discharge).

One of the most recent industrial applications of LTP is large-format UV–digital printing [13,14,15]. The stability of such printing depends on the choice of substrate materials and the printed color pattern. Commonly used polymeric materials have a smooth surface and low surface energy, which often (but not inevitably) causes an insufficient adhesion of inks used for UV–digital printing [15,16,17,18]. The adhesion of the printed color pattern, or its abrasion resistance, then affects the final product’s usage and positioning, or the printed substrate’s impact, depending on climatic conditions (e.g., outdoor conditions, humidity, and temperature).

This paper used the LTP treatment by special diffuse coplanar surface barrier discharge (DCSBD) [10]. To date, the DCSBD has been investigated for the surface treatment and functionalization of various polymers [11,12,19] and biomaterials [20,21]. Its compact and optimized electrode geometry, together with the optimized supplied voltage parameters (f~15 kHz, U_p-p_~20 kV), provides high-power-density, macroscopically homogeneous plasma due to a high presence of diffuse plasma and ensures fast and homogenous surface treatment [22,23]. Moreover, the reliability and scalability of DCSBD plasma operation tested in various gases, humidities, and environments makes this technology especially suitable for industrial-scale in-line material processing.

In this study, the DCSBD plasma modified the surface of polymethylmethacrylate (PMMA) and polycarbonate (PC) substrates, often considered glass substitutes in UV–digital printing. The end-user of UV–digital printers requires a substantial increase in the adhesion and standardization of the printed layer quality on polymer substrates from different manufacturers [18]. Thus, process variables (plasma reactor configuration, treatment time, and treatment distance) were identified and used for experimental tests. The polymer surface’s chemical composition and morphology changes are studied. The adhesion properties of the activated samples are determined by peel-force measurement and a standard crosscut ink adhesion test after UV–digital printing.

## 2. Materials and Methods

### 2.1. Materials

The industrial-grade PC and PMMA substrates provided by Effetec s.r.o. (Čelechovice na Hané, Czech Republic) were used for experiments. Samples were delivered in the form of boards with dimensions of 15 × 10 cm^2^ with an approximate thickness of 3 mm. For different experiments, samples were cut into smaller pieces with dimensions of 1.5 × 10 cm^2^. Protective foil was removed prior to plasma treatment without any additional cleaning of the sample surface.

### 2.2. Plasma Treatment

The LTP treatment of polymeric substrates proceeded at different distances from the sample (225 μm, 300 μm, and 450 μm) in the dynamic mode (i.e., plasma source moving above the substrate). The exposure time varied in the range of 1–5 s. The short exposure times were ensured by changing the speed of plasma source movement, and prolonged treatment resulted from a repeated 5 s exposure. Plasma-treated samples were stored for up to 3 days in a desiccator cabinet at 23 °C and a humidity of 40% to ensure the ageing at controlled conditions.

The plasma was generated by two dielectric barrier discharge versions, DCSBD and half-DCSBD (HDCSBD). The DCSBD generates a macroscopically homogeneous LTP with an input power of 400 W at ~15 kHz [23]. The active-discharge area is ~8 × 20 cm^2^. Figure 1a shows the simplified scheme of the DCSBD plasma source. The more detailed technical specifications of DCSBD are listed in [10].

The HDCSBD is a unique approach to the DCSBD configuration manufactured by KYOCERA Inc. (Kyoto, Japan) but designed and protected by Masaryk University. The cooling by flowing dielectric oil is replaced with an industrial aluminum heat sink and a fan with adjustable airflow. The scheme of HDCSBD is shown in Figure 1b. Plasma generated by HDCSBD has comparable properties to plasma created by DCSBD. The input power is 200 W at ~20 kHz, and the active discharge area is ~8 × 10 cm^2^. These parameters served to design the prototype of the printing device with the implemented plasma technology photographed in Figure 2.

### 2.3. Analytical Methods for Plasma Treatment Evaluation

#### 2.3.1. Wettability Measurements

To calculate dispersive and polar components of surface free energy (SFE), the droplets formed on the substrate were evaluated by the Owens–Wendt regression model [24] using three liquid methods (deionized water, 99% diiodomethane, and 99.8% ethylene glycol from Sigma-Aldrich (St. Louis, MO, USA)). An average of 14 droplets of each liquid placed by a micropipette with a constant volume of 1 µL were used to estimate the contact angles. To capture the photos and measure the profile of the droplets, we used a See (Surface Energy Evaluation) System analyzer (Advex Instruments, s.r.o., Brno, Czech Republic). The standard error in SFE measurement was approx. 1 mJ/m^2^.

#### 2.3.2. Adhesion of Surface

The static material testing machine Texture Analyser TA.XT plusC (Stable Micro Systems, Surrey, UK) measured the samples’ adhesive properties, equipped with a load cell of a 50 N range. According to FINAT method no. 1, the adhesion test (a 180° adhesion peel test) was performed on the samples. The loading speed was 10 mm per minute. The preparation of the samples consisted of attaching a special adhesive tape (TQC Sheen, Rotterdam, the Netherlands) according to ISO 2409:1999 [25] and ensuring 12 passes over a taped area with a pre-calibrated rolling pin (2 kg). For average peel-force calculation, the peel test on a length of 90 mm was repeated on at least 5 samples. The whole process was captured by the software Exponent Connect version 7,0,6,0.

#### 2.3.3. Chemical Characterization

The chemical changes on the surface were analyzed by X-ray photoelectron spectroscopy (XPS) performed by the spectrometer Axis Supra (Kratos Analytical Ltd., Manchester, UK) under the same conditions as in [11]. A monochromatic AlKα energy radiation of 1486.6 eV emitted photoelectrons from a 300 × 700 µm^2^ sample area perpendicular to a used analyzer. During the analysis, a charge neutralizer electron source compensated for the polymer sample charging. Energy calibration was performed by shifting spectra according to a reference peak. Survey spectra were collected using an analyzer pass energy of 80 eV and high-resolution spectra for pass energy of 20 eV with a step size of 0.1 eV. The calibration of spectra, further processing, and standard fitting were carried out using CASA software (trial version CasaXPS 2.3.16, CASA international nv, Olen, Belgium).

#### 2.3.4. Morphological Characterization

The roughness changes after the treatment were measured using atomic force microscope AFM NTEGRA Prima (NT-MDT, Moscow, Russia) in a semi-contact mode using golden-silicon probes AN-NSG10 (Applied NanoStructures Inc., Mountain View, CA, USA) with a typical force constant of 11.8 N/m and typical resonant frequency of 240 kHz. The Root Mean Square (RMS) and Average roughness were estimated from 100–400 μm^2^ with a resolution of 512 × 512 px^2^ and a scanning frequency of up to 1 Hz.

The morphology changes were observed by scanning electron microscope SEM Mira3 (Tescan, Brno, Czech Republic). The detector of secondary electrons and accelerating voltage of 7–10 kV were used to magnify the surfaces up to 50,000 times. Before measurement, both substrates were coated with 20 nm of the Au/Pd layer by sputter coater Quorum Q150R-ES (Quorum Technologies, Lewes, UK).

#### 2.3.5. UV–Digital Printing and Ink Adhesion Evaluation

The custom UV LED head (395 nm, 14 W/cm) cured the acrylate-based ink (EFFE811, Effetec s.r.o.) and pattern with 360 × 1200 DPI resolution. Testing set CC3000 (TQC-Sheen, Rotterdam, The Netherlands) was used to perform crosscut and tape tests according to ISO 2409:2007 [26] to evaluate the ink adhesion right after and 24 h after UV–digital printing. After the crosscut peel test, the numbers and boundaries of cuts were examined and compared to the untreated substrate. For this standard test, the samples kept their dimensions of 15 × 10 cm^2^, and patterns of different colors were printed by a prototype of a UV–digital printing machine (Effetec s.r.o) with implemented HDCSBD technology (Figure 2).

## 3. Results and Discussion

The authors will present the results of the DCSBD and HDCSBD treatment of transparent polymers in the following sections. The following results should compare and prove the compatibility of used plasma sources. Furthermore, the authors intended to emphasize the choice of HDCSBD for its implementation in the UV–digital printing process.

### 3.1. Wettability and Adhesion of LTP-Treated Substrates

The polymers’ total surface free energy (SFE), dispersion, and polar components were calculated for a better understanding of the wetting changes in plasma-treated substrates. For better comparison, the PMMA and PC substrates were treated at different distances and times by both DCSBD and HDCSBD sources. Figure 3 and Figure 4 show that the total SFE increased after short LTP exposure, implying improved wettability of polymer surfaces.

Right after the LTP treatment of PMMA, the polar part increase occurred. After 1-day ageing, the polar part slightly decreased but did not return to the initial value. After 3-day ageing, the achieved polar part remained mostly unchanged within the error. However, the polar part of samples treated for 1 s by DCSBD and HDCSBD at 300 and 450 μm decreased faster during storage.

The dispersion component of PMMA increased slightly by max. 3 mJ/m^2^ after the LTP treatment, except for HDCSBD treatment at 300 μm. However, after ageing, the dispersive part of all samples tended to recover toward a slightly increased value compared to the reference value.

In agreement with the study of Homola et al. [27], PMMA from different suppliers and sonicated by isopropanol achieved similar polar values after the 1 s treatment compared with 3 and 9 s air DCSBD exposure at a 300 μm distance. However, this value (13.1 mJ/m^2^) was unstable and decreased to 7.7 mJ/m^2^ after 3-day ageing. The dispersive part increased more than in our experiment, contributing to a higher increase in total SFE.

In the case of LTP treatment of PC, the polar component significantly increased by 17–22 mJ/m^2^ while the dispersive part decreased by 4–9 mJ/m^2^. After ageing, the polar component decreased in time while the dispersive part was recovering. However, the SFE components did not achieve initial values even after 3-day ageing.

In [28], industrial-grade PC was treated by DCSBD at a distance of 300 μm. Due to hydrophobic recovery, the gained total SFE decreased by 8–9 mJ/m^2^ after 3 days. In our experiment, the samples treated by DCSBD and HDCSBD at the same distance decreased their total SFE by 5–7 mJ/m^2^ and 5–6 mJ/m^2^ after 3 days, respectively.

Incorporating polar functional groups often explains the increase in the polar component on tested polymers. These can be introduced onto the surface due to generating reactive oxygen and nitrogen species (RONS) in air plasma. Moreover, the air humidity and polymer chain characteristics mediate hydroxyl and carboxyl group formation [7,9,11]. The dispersive component reflects the van der Waals bonds and varies due to the incorporation of nonpolar functional groups (e.g., methyl) and changes in surface morphology. This behavior is also described in the literature [29] and, in our case, agrees with the XPS results (Section 3.2).

Surface activation with LTP generally modifies the SFE or surface reactivity through surface oxidation and chemical grafting [5,30,31]. The presented results of SFE measurement imply the formation of polar oxygen-containing functional groups, resulting in increased surface wetting and possibly providing improvement in adhesion [12,32,33]. Studies have also shown the influence of surface roughness on the coating adhesion after plasma activation. These two complementary effects can increase adhesion phenomena at the interface between the coating and substrate [8,30,33,34]. Nevertheless, as mentioned later (Section 3.3 and Section 3.4), the roughness changes could not dominantly contribute to ink adhesion in the case of both tested polymers.

The adhesion changes on PMMA and PC substrates after air DCSBD and HDCSBD treatment at different distances are shown in Figure 5 and Figure 6. The 1 s treatment was excluded from the adhesion evaluation due to many samples and measurements needed right after exposure and after 24 h ageing. Other reasons were the aforementioned fast ageing of 1-s-treated PMMA samples and optimal treatment time evaluated in previous studies [27,28] on similar industrial-grade substrates.

After the DCSBD treatment of the PMMA substrate, the adhesion increased, while the 5 s treatment at 225 and 300 μm distances showed slightly better performance and peel force at 0.27 N/mm. In the case of HDCSBD, higher peel-force values (0.26–0.34 N/mm) were achieved after treatment. In this case, prolonged exposure time improved the adhesion, while the distance of the treated substrate had the opposite effect. These tendencies are evaluated within the error. Moreover, it is worth mentioning that lower measurement error (i.e., 5 s treatment) often indicates the improved homogeneity of large surface treatment [11,19,28]. These results also correlate with SFE measurements.

The peel-force measurements on the PC substrate showed increased adhesion after the LTP treatment. The substrate adhesion almost doubled in the case of DCSBD at a distance of 225 μm. Increasing the distance of the plasma source to the substrate resulted in decreased peel force. The exception was the 5 s HDCSBD treatment at 450 μm, where the surface achieved a comparable peel force of 0.31 N/mm to PC treated at 225 μm. In the case of PC, both sources improved surface adhesion performance (average value and/or error ~homogeneity) after 5 s treatment.

The measuring of PMMA and PC samples stored for 24 h revealed a dramatic decrease in achieved adhesion, most visible at 225-um-distance treatment. Functional groups that aid adhesion at the polymer’s substrates include C=O, CO, COO, OH, and –OOH. The uptake of environmental contaminants, re-orientation of surface groups, and further chemical reactions result in an ageing effect. Therefore, the surface’s wettability is recovered with time, thus worsening its adhesion [2,35]. Nevertheless, the studied application of DCSBD and HDCSBD plasma sources in the in-line process should diminish the ageing of pre-treatment.

For further analyses, i.e., XPS, SEM, and AFM, the treatment at a 225 μm distance was excluded due to the high sample amount and data processing time. This decision was supported by the decrease in achieved adhesion after ageing. The decision on further analyses also reflects the studied implementation of plasma pre-treatment into the UV-printing in-line process. Whereas the thickness of an industrial-grade transparent polymer substrate can vary on large-area plates, the consequential damage (scratching, etc.) could result in the defect of a final product.

### 3.2. Chemical Changes on LTP-Treated Substrates

The XPS analysis of LTP-treated transparent polymers revealed chemical changes on both substrates. The elemental composition and functional group concentration on PMMA and PC surfaces before and after plasma treatment are listed in Table 1 and Table 2, respectively.

The atomic concentration analysis focused on the main PMMA- and PC-composing elements, carbon and oxygen, and on nitrogen contained in air. Despite the plasma treatment in air, the atomic concentration of N 1*s* increased negligibly after the treatment. The tendencies of increasing concentration due to the time or distance of the treatment are not obvious (within the error). The other element concentrations (e.g., Si 2*p*) are not listed in Table 1 and Table 2, as they are not expected on pristine samples and implied surface contamination.

After the treatment, the C and O concentration changes are obvious and compared as the O/C ratio. On the PMMA surface, after the 3 s plasma treatment at 300 and 450 μm distances, the O/C ratio increased from the initial 0.26 to 0.29 and 0.36, respectively. However, after prolonged treatment (5 s), the O/C ratio reduced to the initial value at 300 μm and decreased even more to 0.20 with increased distance.

The C–C/C–H concentration increase after 5 s treatment indicates cross-linking on the surface induced by prolonged plasma treatment and UV emitted by the plasma [6]. The changes in wettability and surface free energy on PMMA were associated with a pronounced increase in polar components. The C–O group contributing to that component increased after the treatment. The creation of C–O and the loss of other oxygen-containing groups imply their conversion to low-weight volatile fragments (e.g., CO and CO_2_) during the LTP interaction [31], contributing to cross-linking.

Similarly to PMMA, the initial O/C ratio on the PC surface (0.13) increased after the plasma treatment by a value of 0.08–0.18. The O/C ratio change was more pronounced for a shorter distance. Even though the authors of [38] specified the effective treatment distance as 300 μm, the interaction of the surface and plasma also occurs at a higher distance. The UV-resistant PC treated at 450 μm needed more time to increase oxygen concentration to a similar value as in the case of 300 μm. This effect is probably connected to a more remote plasma effect, e.g., gaseous chemical products and long-living particles.

The increase in polar components on the PC surface can be explained by creating polar oxygen-containing functional groups. The deconvolution of C 1*s* high-resolution spectra revealed new C=O groups and shake-up-satellite π–π*. The increase in C=O and O–C=O concentration correlated with the changes in the O/C ratio and highlighted a remote plasma effect. These groups contribute profoundly to the polar component of PC SFE. The other oxygen-containing groups decreased after the 3 s plasma treatment but returned to their initial value after the 5 s treatment.

Compared with PMMA, we can assume that cross-linking does not play a role in PC chemical surface changes. Possible hydrogen abstraction from the polymer chain and reduction in nonpolar groups (e.g., methyl) on the surface contributed to C–C/C–H concentration reduction [9,11]. The changes in functional groups are most probably connected with oxygen and nitrogen group incorporation from air and gaseous plasma products. Similar tendencies were observed after air and oxygen plasma treatment [7,39]. The authors of [39] assumed that initial sidechain and ring oxidation predominated upon longer plasma treatment.

### 3.3. Morphological Changes on LTP-Treated Substrates

The polymer–plasma interaction can often cause the etching and heat–UV degradation of the surface. However, such effects are not expected after the short LTP treatment. We analyzed the surface roughness and morphological changes using AFM and SEM to confirm the suitability of the plasma and experimental conditions. The roughness values are listed in Table 3 for pristine and HDCSBD-treated PMMA and PC substrates.

Prolonging the plasma treatment time resulted in an increase in surface roughness on both materials. A similar trend was observed in roughness after shortening the plasma treatment distance. The maximum surface changes were recorded for both substrates after 5 s of plasma exposition at a distance of 300 μm.

The roughness values changed more significantly in the case of PMMA. The surface roughness increased approx. three and two times after treatment at 300 and 450 μm, respectively. In the case of PC, after 5 s treatment, the observed surface roughness increased 2.5 and 1.2 times for the abovementioned distances, respectively.

The previous works described the effect of treatment time on roughness change in correlation with the SFE and contact angle on PMMA and PC [27,28]. As a result, the chosen optimal treatment times correspond with our study. Moreover, in previous work, the 30 s plasma over-exposure of PMMA led to a higher surface roughness value. This increase in the RMS roughness value was explained by thermal damage caused by sample overheating. However, the authors of [40] ascribe the five-time increase in nanoscale roughness to UV–ozone-induced PMMA polymer chain scission and local melting after 5 min of UV–ozone exposure. Thus, a further increase in roughness can be expected after a prolonged exposition of plasma and its components (e.g., UV and ozone) [3,27].

Following the roughness measurement results, plasma-induced morphological changes are not verifiable from SEM micrographs (Figure 7). After removing the protective tape, dust particles slightly contaminated the substrates due to the created static charge on the substrate. Moreover, the pristine substrates are not perfectly smooth, which suits the AFM results, whereas the pristine PMMA roughness is higher than in the case of PC. The small cracks visible on the micrographs are Au/Pd conductive coating ruptures after the high-energy electron impact at the high-resolution mode.

After the LTP plasma treatment at a distance of 300 μm, SEM revealed the contamination and roughness changes on the surface of PMMA. The PMMA sample treated for 5 s showed visible “bubble-like” structures, implying physicochemical etching of the upper surface, whereas the 3-s-treated sample was mostly contaminated with smaller similar structures. After prolonged treatment, we can assume the effect of “bubble” growing by aggregating these smaller structures. Roughness changes are less obvious after treatment at 450 μm. Furthermore, the aforementioned structure creation was not apparent from SEM micrographs.

The PC sample LTP treated for 3 s and 5 s at a 300 μm distance revealed roughness changes on a part of the surface in comparison to other PC samples. The “swollen” areas were created after 3 s and joined after longer treatment to a larger area. This effect was not visible at a distance of 450 μm, and small visible particles indicate contamination from ambient air.

Similarly to the results of chemical characterization, more pronounced changes on both substrates were observed for the LTP treatment at a distance of 300 μm. However, compared to our previous work and works using oxidizing plasma [3,19,28,37], these changes in roughness and morphology are insignificant, and we did not expect their dominant contribution to ink adhesion. After the LTP treatment, no visible changes were observed in the transparency of the substrates.

### 3.4. Ink Adhesion of UV–Digital Printing on LTP-Treated Transparent Polymer Substrates

Polymer materials are produced worldwide by different suppliers with slightly different production processes. The end-user of UV–digital printers requires a substantial increase in the adhesion of the printed layer and standardization of the printed layer quality [16]. To achieve the best possible results and implementation of an LTP plasma source for flat surface treatment prior to the UV–digital printing, the authors constructed and further optimized the plasma unit introduced in Figure 1b. Due to its robust and compact construction, this HDCSBD was selected as the appropriate plasma source for the in-line UV-printing machine (Figure 2). The optimum conditions for effective large-area UV–digital printing during the in-line testing were set to the distance of 300 μm and at treatment times of 1, 3, and 5 s for both substrates.

The ink hardened by the LED UV lamps is shown in Figure 8, Figure 9, Figure 10, Figure 11 and Figure 12. The crosscut and tape test qualitatively evaluated the ink-to-substrate bond. All printed patterns presented proper adhesion to the surface of both materials. The differences between untreated and treated surfaces were more visible on dark patterns, consisting of more color pigments (Figure 8). Thus, we compared the margin of cuts and possible ink shrinkage instead of ISO classification.

After the short plasma treatment of PMMA (Figure 9), the crosscut and tape test revealed a slight improvement in UV–ink adhesion. However, the margin distortion of the cut after 24 h and ink shrinkage (Figure 10) are not substantially reduced on treated PMMA compared to the untreated sample. In this case, UV–ink had more time to cure and interact with the LTP-treated substrate [16]. Thus, the cut margins do not show a plain dependence of ink adhesion quality on plasma treatment time.

The change in SFE of the treated surface affects the ink droplet shape and its diameter, which can develop in time. This effect is more pronounced for LTP-treated substrates [18]. Moreover, these authors observed that the UV–ink could undergo considerable degradation of its adhesion after 24 h on untreated and cleaned surfaces compared to stable ink adhesion to LTP-treated surfaces.

In our case, we did not observe ink adhesion degradation even on untreated PMMA. From the comparison of treated samples cut right after and after 24 h, we can deduce the ink adhesion improvement due to ink curing and chemical interaction with the functional group created after LTP treatment.

In the case of the PC substrate, the short LTP treatment improved the ink adhesion immediately and 24 h after printing (Figure 11 and Figure 12). Moreover, the roughness change on the PC surface (Section 3.3) is not considered dominant in affecting the UV–ink adhesion. The authors link better ink adhesion with the SFE increase, adhesion improvement, and chemical changes on the LTP-treated PC substrate [33,35].

## 4. Conclusions

This paper introduces a special LTP type for flat PMMA and PC substrate pre-treatment in UV–digital printing applications. The distance range (225–450 μm) of the substrate to the plasma source was chosen to fit the requirements of an in-line industrial printing scale. The short (1–5 s) treatment using DCSBD and HDCSBD plasma sources enhanced the surface properties comparably for tested PMMA and PC materials. The activation of substrates was steady within 72 h, and none of the treated substrates showed full hydrophobic recovery. Since plasma introduced oxygen-containing polar groups onto the surface, the SFE increase is ascribed to the polar component. No significant variation is seen for the dispersive component, which agrees with the XPS results. The XPS analyses revealed the increase in the O/C ratio and implementation of polar functional groups onto both polymer substrates, namely C–O in the case of PMMA and C=O/O–C=O in the case of PC. The morphological changes were minimal within the plasma treatment times and distances used, not affecting the quality and transparency of the material. Minimal morphological changes were observed on SEM micrographs, corresponding with a slight roughness increase on both substrates revealed by AFM.

Even 1 s treatment substantially increased ink adhesion and UV–digital printing quality for PC substrates. For resistant substrates, such as PMMA, longer treatment times were needed. However, improved ink resistance against delamination was achieved on plasma-treated substrates in the vicinity of the cut. According to this study, the authors ascribe the ink adhesion improvement to the chemical changes (SFE, XPS) rather than to the nanoscale roughening of the plasma-treated substrates.

Implementing tested LTP sources into the UV–digital printing process improves ink adhesion after short treatment of PMMA and PC substrates. Moreover, such a process lacks the drawbacks of different LTP sources and substitutes the additional need for ink’s surface tension control and wet chemical pre-treatment by so-called primers.

## 5. Patents

The legally protected outcome and certified technology resulting from the work reported in this manuscript are the Czech utility model “A device for plasma modification of large-area planar substrate before UV-digital printing” (no. 35867, PUV2022-39679) and certified technology “Technology for ink adhesion improvement of large-area printing on glass and polymer substrates by atmospheric pressure plasma activation” available at https://www.isvavai.cz/riv (accessed on 28 April 2023).

## Figures and Tables

**Figure 1 nanomaterials-14-00426-f001:**
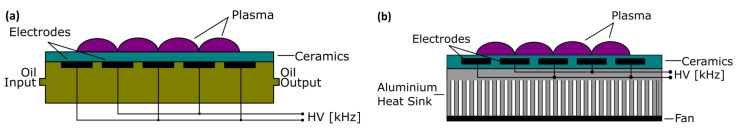
The simplified scheme of (**a**) DCSBD and (**b**) HDCSBD plasma technology.

**Figure 2 nanomaterials-14-00426-f002:**
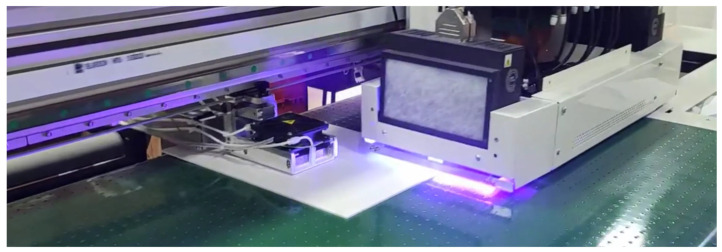
The photograph of the UV–digital printer prototype with implemented HDCSBD technology during UV-LED curing (legally protected by outcomes listed in Section 5).

**Figure 3 nanomaterials-14-00426-f003:**
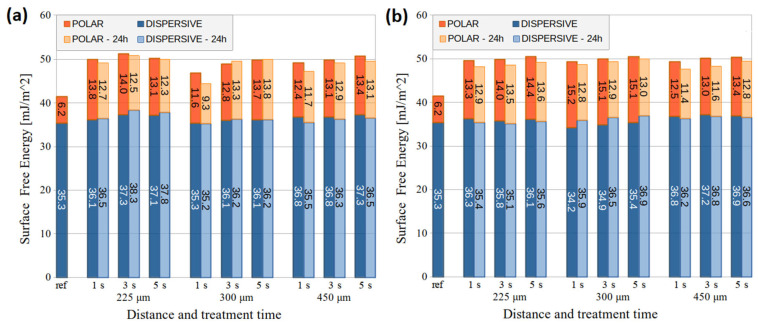
The SFE (determined from the average angle of 14 droplets per liquid) of PMMA substrates treated by (**a**) the DCSBD and (**b**) the HDCSBD plasma technology with standard error ~1 mJ/m^2^.

**Figure 4 nanomaterials-14-00426-f004:**
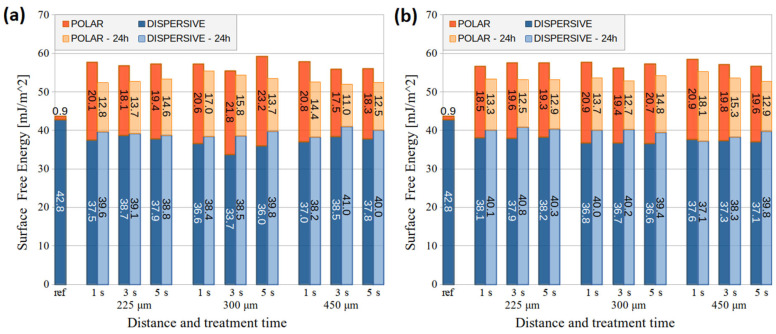
The SFE (determined from the average angle of 14 droplets per liquid) of PC substrates treated by (**a**) the DCSBD and (**b**) the HDCSBD plasma technology with standard error ~1 mJ/m^2^.

**Figure 5 nanomaterials-14-00426-f005:**
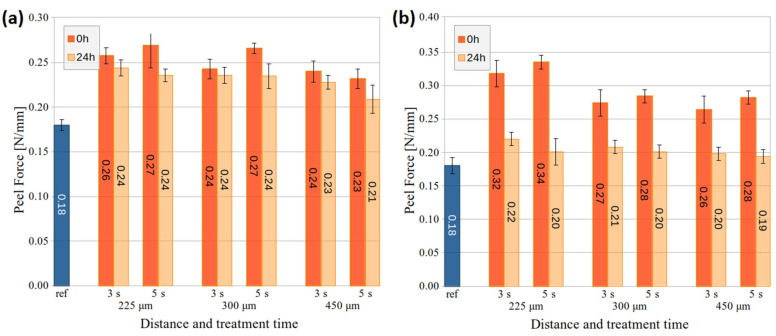
The peel force of PMMA substrates treated by (**a**) the DCSBD and (**b**) the HDCSBD plasma technology estimated from measuring a minimum of 5 samples for each condition.

**Figure 6 nanomaterials-14-00426-f006:**
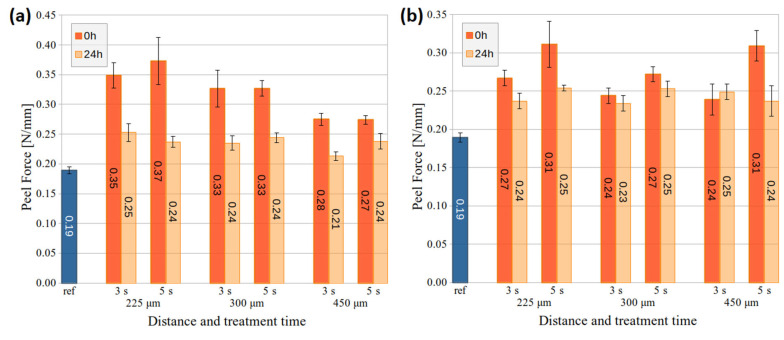
The peel force of PC substrates treated by (**a**) the DCSBD and (**b**) the HDCSBD plasma technology estimated from measuring a minimum of 5 samples for each condition.

**Figure 7 nanomaterials-14-00426-f007:**
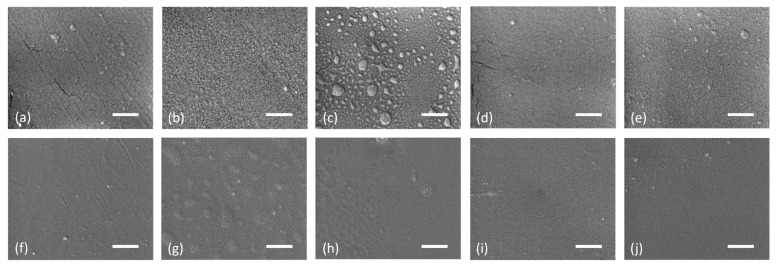
SEM images of PMMA (**a**–**e**) and PC (**f**,**j**) after different HDCSBD treatment conditions: reference sample (**a**,**f**), treated at 300 μm distance for 3 s (**b**,**g**) and 5 s (**c**,**h**), treated at 450 μm distance for 3 s (**d**,**i**) and 5 s (**e**,**j**) (scale: 1 μm).

**Figure 8 nanomaterials-14-00426-f008:**
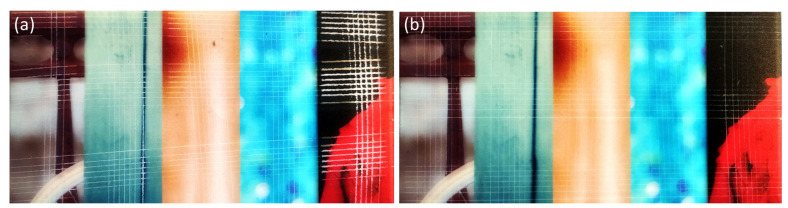
The photographs of color patterns printed on PMMA substrates after the crosscut test: (**a**) reference and (**b**) 5 s treatment.

**Figure 9 nanomaterials-14-00426-f009:**
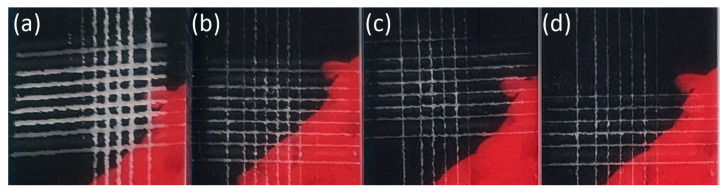
The photograph of the black pattern after the crosscut and tape test right after printing on PMMA substrate treated for (**a**) 0 s, (**b**) 1 s, (**c**) 3 s, and (**d**) 5 s.

**Figure 10 nanomaterials-14-00426-f010:**
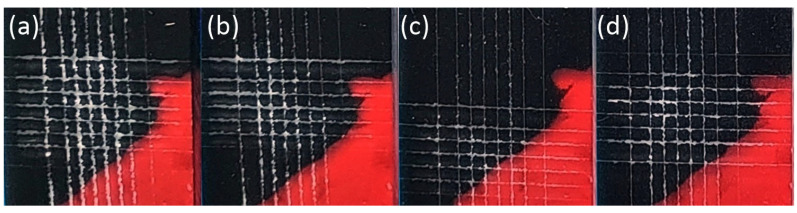
The photograph of the black pattern after the crosscut and tape test after 24 h of printing on PMMA substrate treated for (**a**) 0 s, (**b**) 1 s, (**c**) 3 s, and (**d**) 5 s.

**Figure 11 nanomaterials-14-00426-f011:**
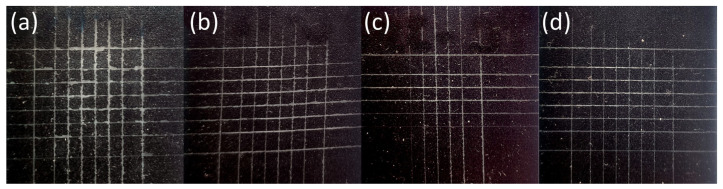
Detail of the black pattern on PC samples after the crosscut and tape test right after the printing: (**a**) reference and treated for (**b**) 1 s, (**c**) 3 s, and (**d**) 5 s.

**Figure 12 nanomaterials-14-00426-f012:**
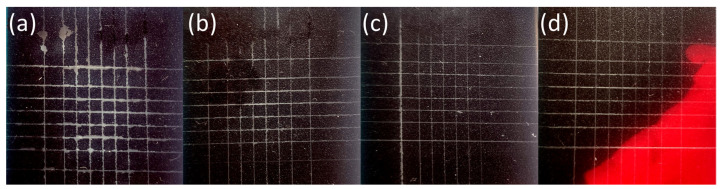
Detail of the black pattern on PC samples after the crosscut and tape test 24 h after the printing: (**a**) reference and treated for (**b**) 1 s, (**c**) 3 s, and (**d**) 5 s.

**Table 1 nanomaterials-14-00426-t001:** The atomic concentrations and the relative areas of the C 1*s* peak components of the PMMA surface analyzed by XPS measurement after the treatment by HDCSBD.

Treatment [um]		At. Conc. [%] ^1^				Functional Group Conc. [%] ^2^			
		C	O	N	O/C Ratio	C–C/C–H	C–O	C–C=O	O–C=O
						284.8 eV	286.7 eV	285.5 eV	288.8 eV
REF	0 s	79	20	-	0.26	52	12	22	15
300	3 s	76	22	1.6	0.29	51	19	18	12
	5 s	76	19	1.6	0.26	65	19	8	8
450	3 s	71	26	2.3	0.36	53	25	9	14
	5 s	81	16	1.6	0.20	66.5	16	10	7

^1^ Estimated from survey spectra. ^2^ Estimated by deconvolution of C 1*s* high-resolution spectra [36].

**Table 2 nanomaterials-14-00426-t002:** The atomic concentrations and the relative areas of C 1*s* peak components of the PC surface analyzed by XPS measurement after the treatment by HDCSBD.

Treatment [um]		At. Conc. [%] ^1^				Functional Group Conc. [%] ^2^					
		C	O	N	O/C Ratio	C–C/C–H	C–O	C=O	O–C=O	O–C=(O)_2_	π–π*
						284.8 eV	286.3 eV	287.8 eV	289.1 eV	290.6 eV	292.2 eV
REF	0 s	88	11	-	0.13	71.4	24	-	1	3	-
300	3 s	74	23	1.8	0.31	66	24	2	7	2	-
	5 s	76	22	1.5	0.29	65	23.4	4	4.5	3	1
450	3 s	81	17	1.0	0.21	74	20	2	3	2	-
	5 s	77	21	1.3	0.28	66	24	2.4	3.5	3.5	2

^1^ Estimated from survey spectra. ^2^ Estimated by deconvolution of C 1*s* high-resolution spectra [37].

**Table 3 nanomaterials-14-00426-t003:** The roughness of PMMA and PC after HDCSBD plasma treatment with a relative standard error of 5–10%.

PMMA				PC			
Treatment		Roughness [nm]		Treatment		Roughness [nm]	
		RMS	Average			RMS	Average
REF	0 s	3.3	2.6	REF	0 s	0.57	0.31
300 μm	3 s	8.8	6.5	300 μm	3 s	0.52	0.41
	5 s	9.95	7.7		5 s	1.40	0.68
450 μm	3 s	5.5	2.9	450 μm	3 s	0.45	0.37
	5 s	7.0	4.9		5 s	0.68	0.38

## Data Availability

All the necessary data are contained within the article.

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
