# Peer review of "Plasma Treatment of Large-Area Polymer Substrates for the Enhanced Adhesion of UV–Digital Printing"

_nanomaterials, 2024, doi:10.3390/nano14050426_

Round 1
Reviewer 1 Report
Comments and Suggestions for Authors
The research paper reports an experimental study on the use of low-temperature plasma (LTP) for the treatment of large-area polymer substrates. This work focused on an industrial application for UV-digital printing. The authors also show a prototype for the validation of the main idea. The LTP treatment is composed of a special diffuse coplanar surface barrier discharge (DCSBD), as explained in Ref 9. There is sufficient characterization presented with well-explained results, supported calculations, and detailed procedures. I consider the research interesting for the journal's potential readers, and I recommend its publication with minor changes:
- Please, include the number of samples and the statistical analyses performed not only in the materials and methods but also in Figures 3,4,5, and 6.
Comments on the Quality of English LanguageNo additional comments.
Author Response
The authors would like to sincerely thank the reviewer for the time spent reading the manuscript, the thorough revision, and suggestions for improving our manuscript. The authors added the requested information to the caption of Figures 3-6.
Reviewer 2 Report
Comments and Suggestions for Authors
In this manuscript, Michal et al reported the plasma treatment of large are polymer substrates for enhanced adhesion of UV-digital printing, the paper can be accepted after the following issue were concerned.
1. The diffuse coplanar surface barrier discharge(DCSBD) is a very novel compact configuration application and is used here. The authors should briefly introducte DCSBD and the introduction, especially its main contribution on the surface field.
2. The surface roughness plays an important role here, however, will the LTP treatment time affect the results? More discussion should be added.
3. For AFM roughness, what is the pressure force for the scanning?
Author Response
The authors would like to sincerely thank the reviewer for the time spent reading the manuscript, the thorough revision, and suggestions for improving our manuscript.
The authors concerned with specified issues in the manuscript and added the answers and comments:
- Thanks for this suggestion. The authors added a paragraph to the Introduction with more information about DCSBD technology. We believe that such an expanded Introduction section will provide the reader with a more comprehensive view of the issue studied in the article (L50-57).
-
The authors thank the reviewer for this comment. The authors added a further discussion to the manuscript text (L322-330).
In our previous works cited in the manuscript, the study of DCSBD air plasma effect was studied on PMMA and PC substrates. The effect of treatment time on roughness change was described in correlation with SFE and water contact angle (WCA) studied for treatment times 1-10 s for PMMA and PC. As a result, the chosen optimal treatment times and roughness change analyses correspond with our study. Moreover, in previous work, the 30 s plasma over-exposure of PMMA led to a considerably higher surface roughness value. This increase in RMS roughness value was explained by thermal damage caused by the overheating of a sample. In different work, the authors ascribe the 5x increase of nanoscale roughness to UV-ozone induced PMMA polymer chain scission and local melting after 5 mins of UV-ozone exposure (Liu J. et al. 2016). However, no time dependence was discussed.
- In the semicontact mode and for the NSG10 probe, the typical force constant is 11.8 N/m, and the typical resonant frequency of 240 kHz is used. The authors added this information to the Materials and Methods.